# A sampling-based circuit for optimal decision making

**Camille E. Rullán Buxó**
Center for Neural Science
New York University
New York, NY 10003
ch2880@nyu.edu

**Cristina Savin**
Center for Neural Science
Center for Data Science
New York University
New York, NY 10003
csavin@nyu.edu

## Abstract

Many features of human and animal behavior can be understood in the framework of Bayesian inference and optimal decision making, but the biological substrate of such processes is not fully understood. Neural sampling provides a flexible code for probabilistic inference in high dimensions and explains key features of sensory responses under experimental manipulations of uncertainty. However, since it encodes uncertainty implicitly, across time and neurons, it remains unclear how such representations can be used for decision making. Here we propose a spiking network model that maps neural samples of a task-specific marginal distribution into an instantaneous representation of uncertainty via a procedure inspired by online kernel density estimation, so that its output can be readily used for decision making. Our model is consistent with experimental results at the level of single neurons and populations, and makes predictions for how neural responses and decisions could be modulated by uncertainty and prior biases. More generally, our work brings together conflicting perspectives on probabilistic brain computation.

One of the central questions of perception is how organisms reliably estimate hidden or abstract quantities of interest using noisy and ambiguous sensory information. Almost equally important is representing the reliability of these estimates, especially in complex environments and situations of risk, where the uncertainty associated with a choice may radically change the optimal course of action. From basic functions, such as cue combination or motor control, to higher level cognitive tasks, such as decision making and planning, there is substantial evidence that both humans and animals represent and use uncertainty information to guide their actions [1]. Neural correlates have been identified in a number of regions, including the orbitofrontal cortex [2], the cingulate cortex [3], and the lateral intraparietal area (LIP) [4, 5], but the principles behind how uncertainty is represented in neural circuits to support circuit computations remain hotly debated.

Since behavior has been shown to be Bayes-optimal in many situations [6], the problem of perception can be modelled in the framework of Bayesian inference, whereby an observer combines prior information with current observations according to an internal model to arrive at a posterior estimate of a quantity of interest (the 'latent variable'). Moreover, natural statistics are strongly non-Gaussian and there is evidence that human subjects use varied, non-Gaussian prior representations to support behavior [7, 8], so observers must be able to perform inference flexibly, efficiently and adaptively. This raises two questions: which neural organizations allow for these kinds of computations, and how are the associated neural representations mapped into behaviorally relevant action plans?

Currently there are several theories for how neural activity may represent probability distributions. One large class of models assume that neural responses encode parameters of underlying posterior distributions; this includes probabilistic population codes [9, 10], their predecessors, kernel density estimators and distributional population codes [11, 12], and most recently, distributed distributional codes (DDC) [13]. A second class of models relies on stochasticity in recurrent circuits to approxi-

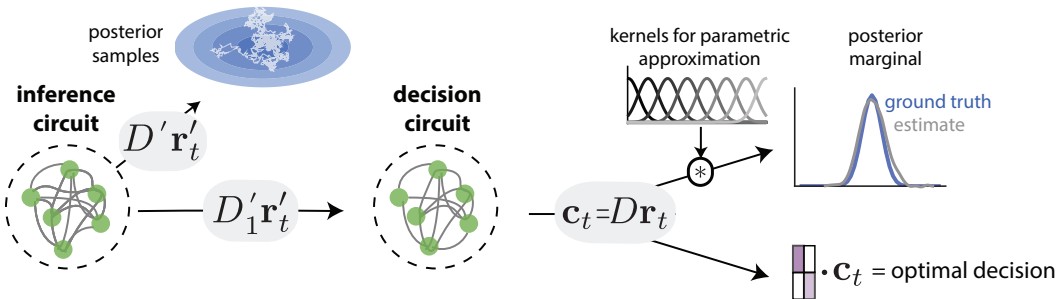

Figure 1: **Schematic of sampling-based optimal decision making in spiking networks.** Approximately inference is performed in a first spiking recurrent network by distributed sampling [22] (example trajectory from joint posterior shown in white). Samples from the task-relevant marginal posterior are read out linearly from this circuit and serve as input to a second network, which integrates them and converts them into a parametric representation. The associated parameters (bank of kernels in gray) are read out linearly from this second circuit and combined with a cost function, reflecting the potentially asymmetric cost of different errors, to generate the final optimal decision.

mately encode probability distributions via sampling [14, 15]. This idea was originally motivated by the practical success of Markov Chain Monte Carlo (MCMC) sampling when performing inference in complex graphical models [14], and had the appeal of being able to flexibly represent complex probability distributions, something that parametric models could not achieve at the time.[1] Sampling has been invoked to explain aspects of perceptual decision making [16], response variability in V1 neurons [17, 18] and structured spontaneous activity in the cortex [19]. Moreover, recent theoretical work has focused on improving the computational efficiency of neural sampling, by increasing sampling speed [20, 21] and improving the robustness of the representation [22]. Nonetheless, sampling-based codes represent uncertainty only implicitly, distributed across time and neurons [22]. How the brain maps neural sampling dynamics into uncertainty-calibrated decisions remains a key open question.

Here, we propose the first spiking circuit for mapping neural samples into appropriate actions. At the core of our idea is the observation that although inference may involve complex high-dimensional posteriors, individual decisions are usually based on low-dimensional marginals of these distributions, which can be represented explicitly using a traditional parametric code. The proposed recurrent network model takes as inputs samples from the task-relevant posterior marginal and integrates them over time in a procedure inspired by online kernel density estimation [23]. We demonstrate the ability of this circuit to perform decision making in several toy examples. We also analyze the spiking activity of the network to look for signatures of uncertainty and probabilistic computation at the level of single neurons and population activity and find that our model recapitulates a variety of empirical observations in neural data. The model also makes predictions for how uncertainty and prior biases affect neural activity and population latent dynamics in this decision circuit.

## 1    Spiking neural network for sampling-based marginalization

In a Bayesian framework, perceptual decision making involves several key computations. First, given a sensory stimulus, $\mathbf{s}$, one needs to compute the posterior over the latent variables, $\mathbf{x}$, that may have given rise to it, $P(\mathbf{x}|\mathbf{s})$. Second, uncertainty about nuisance variables is averaged out, to obtain a marginal distribution over the task-relevant latent $x_i$, $P(x_i|\mathbf{s})$. Finally, this marginal is combined with a task-specific cost function to yield the final decision. The first two steps are straightforward under a sampling framework [22], but parametric representations are needed for the final step to convert the information represented by samples into a spatially and temporally localized code [9].

Our approach uses spike-based distributed sampling for approximate inference and marginalization, then converts the resulting samples into an instantaneous parametric representation that can be used

---

[1]Since recent DDCs are also able to represent complex multidimensional posteriors, the arguments in favor of sampling have moved away from representational flexibility, to its ability to explain many nontrivial features of neural variability in the cortex.

for decision making (Fig.1). For the inference portion of the model, we use the distributed sampling scheme developed by Savin and Deneve [22], implemented using the Poisson version of the encoding model, proposed by Rullán Buxó and Pillow [24]. Briefly, the recurrent circuit embeds Langevin sampling dynamics into a population of spiking neurons so that samples can be read out from the instantaneous neural rates (computed by low-pass filtering the spikes with an exponential kernel), $\mathbf{r}'_t$ via a fixed linear decoder, $\mathbf{D}'$: $\mathbf{x}_t = \mathbf{D}'\mathbf{r}'_t$ (see Suppl. Info. and Refs. [22, 24] for details) [2]. Given these dynamics, samples from the $i$-th marginal are obtained by projecting activity along the axis defined by row $i$ of the decoding matrix, $\mathbf{D}'_i$.

The core of our contribution is the second circuit, which receives inputs $x_{i,t} = \mathbf{D}'_i\mathbf{r}'_t$ from the inference circuit, and constructs a parametric approximation of the associated distribution as a mixture of Gaussian kernels, $\phi_k(\cdot)$, parametrized by kernel weights, $\mathbf{c}$: $\mathrm{P}(x_i) = \sum_k c_k\phi_k(x_i)$. For simplicity, here we assume a collection of $K$ Gaussian basis functions, although alternative choices of kernels are also possible. The parameters $\mathbf{c}$ adapt dynamically with each input sample, following dynamics of the form:

$$\dot{\mathbf{c}}_t = \frac{1}{\alpha}(\boldsymbol{\phi}(x_{it}) - f(\mathbf{c}_{t-1})) \tag{1}$$

where vector $\boldsymbol{\phi}(x_{it})$ concatenates all basis functions evaluated at the current input; function $f(\cdot)$ ensures that the fixed point solution yields a properly normalized distribution. Throughout this paper, we define $f(\vec{\mathbf{c}}_t) = K\langle\phi\rangle\vec{\mathbf{c}}_t = \beta\vec{\mathbf{c}}_t$, such that at the fixed point, the norm is 1, $\sum_k \frac{\phi_k(x_t)}{\beta} = \sum_k c_{k,t} = 1$ (see Suppl. Info. for details). The $\alpha$ term in Eq. 1 can be thought of as a leak, or the discount rate of internal evidence. This term becomes important for studying the dynamics of neural activity and decision making when the marginal posterior is changing, but otherwise does not qualitatively affect our results.

The dynamics implementing the parametric approximation of the marginal are embedded into a spiking neuron population, using the same encoding model used for the first module [24, 25, 27], so that the parameters can themselves be read out from neural activity via another arbitrary, fixed linear decoder, $D$, of size $K \times N$. The communication between circuits involves a linear map along the relevant axis of the inference circuit decoder, followed by point nonlinearities, with a form determined by the kernel $\phi_k(\cdot)$; this map could be implemented by local dendritic non-linearities.

As a simple proof of concept, we first consider inference in a linear Gaussian graphical model (Fig. 2A; see Suppl. Info. for details and simulation parameters), which results in Gaussian posterior marginals. Before stimulus onset, the inference circuit samples from a broad Gaussian prior centered at zero. Once the stimulus is on ($t = 0.15$s), the dynamics switch to sampling from a sharper and correlated posterior (Fig. 2B). We take the first dimension $x_1$ as the decision-relevant latent dimension, with corresponding samples forming the stream of inputs to the decision circuit of size $N = 160$, where they are converted into a parametric representation of the marginal (Fig. 2C). The spiking activity of the neurons in the second circuit reflect these probabilistic computations, both before (spontaneous activity) and after stimulus presentation (Fig. 2D) so that parameters $\mathbf{c}_t$ can be read out from the neural responses (Fig. 2E). They also closely match the ground truth $\sim 150 - 200$ms after stimulus onset, once the dynamics have stabilized (Fig. 2F).

To explore how this representation could be used for optimal decision making, we consider a slightly more involved scenario, in which the task-relevant marginal is a bimodal distribution (formally, a two-component Gaussian mixture with unequal variance, Fig. 3A), a classic example where different choices of cost functions can have very different optimal decisions.[3] For instance the task of estimating $x_1$ has the posterior mean as the optimum, while reporting the most likely value would use a maximum a posteriori (MAP) estimate. Here, we consider a binary categorization task, which requires reporting the sign of $x_1$ as an abstract analogue to binary discrimination (for instance, deciding whether or not an oriented grating is tilted leftwards or rightwards relative to the vertical). The optimal decision in this case is to report the side of the decision boundary that has more mass (integrating out the posterior $\int_0^\infty \mathrm{P}(x_i|\mathbf{s})dx_i$). For our concrete example, there is slightly more mass

---

[2]The decoder weights are pre-specified by the linear dynamical system, consistent with the encoding framework by [25]; however, there exist extensions on how the weights could be learned through local plasticity [26].

[3]Bimodal posteriors are common in probabilistic accounts of bistable percepts [28, 29]

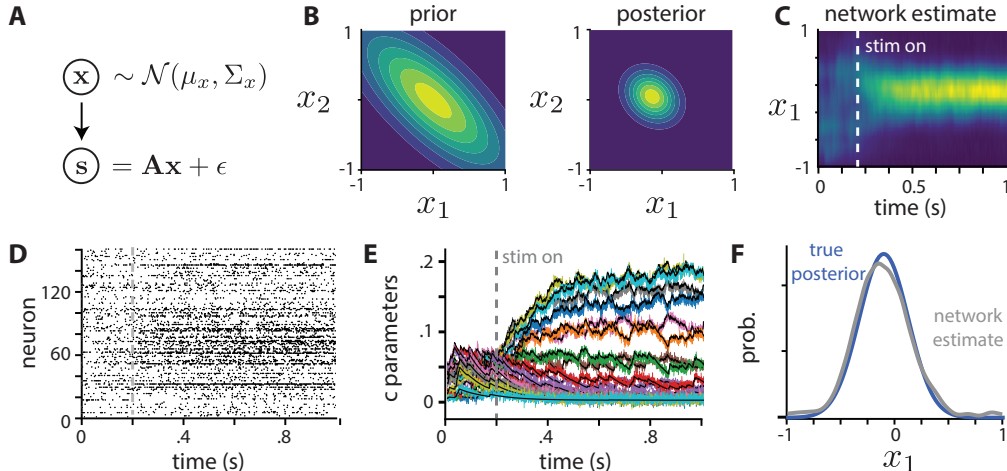

Figure 2: **Decision circuit and proof of concept decision making. A.** Graphical model for our toy example: inference in a linear Gaussian model. **B.** An example prior and posterior with two latent variables. **C.** The time evolution of the parametric approximation of the marginal for $x_1$ of the prior (before stimulus onset), and the posterior (afterward)s. **D.** Spiking activity encoding this information. **E.** Decoded parameters, **c**; ground truth in in black, network estimates in color. **F.** Comparison between estimate and ground truth posterior marginal at the end the trial.

on the negative side, such that with equal costs the optimal decision would be to report 'L'.[4] We also define an asymmetric cost scenario where the cost of mistakenly reporting a negative value is higher than the alternative so the optimal decision changes to 'R'.

The dynamics of the decision circuit are shown in Fig. 3B. In this example, the inference circuit happens to start its trajectory in the rightward mixture component; since Langevin dynamics are slow at crossing regions of low probability [20, 21], initially the decision circuit only 'sees' samples from one component, but eventually both components are explored and the parametric marginal converges to the ground truth. In parallel, a cost-weighted linear projection of the corresponding neural responses tracks the evolution of the estimated cost associated to the two options (Fig. 3C). Indeed, for equal costs option 'L' starts with a high value, as it looks improbable under the initial approximate posterior, with the estimate converging to the correct low value with more samples; comparing the two options yields the optimal decision, 'L'. Similarly, when high costs bias the decision away from this option, the network labels 'R' as the preferred option. Any other choice of cost function can be accommodated by changing the linear readout from the decision circuit. The same is true for estimation tasks based on the posterior mean, which we simulated to reproduce a reduction in the slope of a psychometric function with increasing uncertainty (Fig. 3D) and a shift in due to prior biases (Fig. 3E).

## 2 Neural signatures of sampling-based decision making

The inference and decision circuits share a lot of their underlying structure. Both encode information using the same spiking code; both their responses are driven by samples from the same posterior, directly in the inference circuit, and indirectly — via a linear projection — in the decision circuit. Are the two computations distinguishable at the level of the neural activity?

To investigate the neural implications of our model, we reproduced the experiment used to probe neural correlates of distributed sampling from [22]. Briefly, we assumed that the first circuit performs inference in response to 9 distinct test stimuli, under the assumption of a linear Gaussian graphical model. At the level of the decision circuit, this setup yields corresponding Gaussian marginals with equal variance and means that are evenly spaced on an interval from $-1$ to $1$. We probe the responses

---

[4]Note that codes that only consider the mean and variance of the posterior or ones that only represent the mode (MAP) would yield suboptimal decisions in this example; the correct answer requires knowledge of the full distribution as provided by sampling.

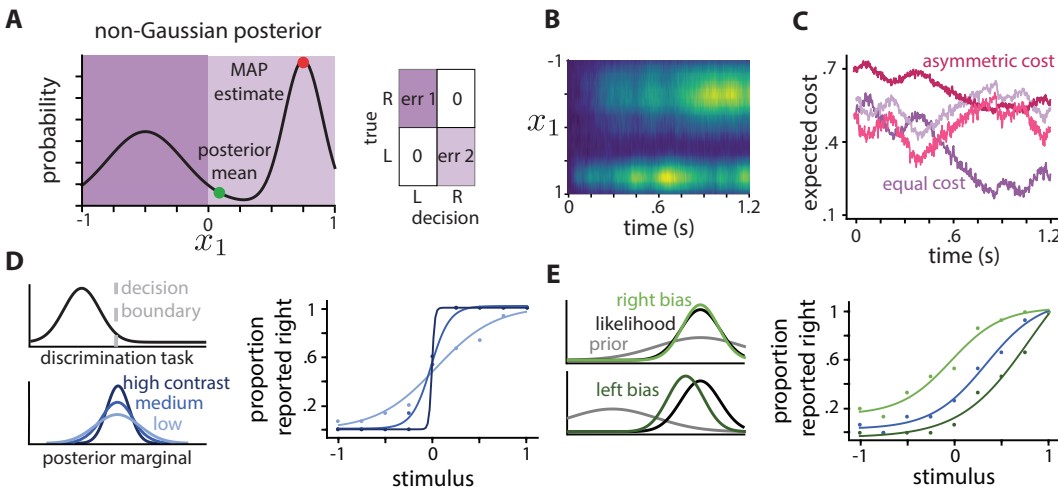

Figure 3: **Toy examples of decision making. A.** A binary decision when the relevant marginal is a two-component Gaussian mixture, so the mean (green) and the mode (red) of the distribution differ. The goal is to optimally report the sign of $x_1$, taking into account any assymmetry in the cost structure. **B.** Evolution of the network's marginal estimate; inference circuit dynamics start in the right posterior mode. **C.** Weighting output by the cost function generates dynamic estimates of the expected cost of the two decisions (L/R) in the case of equal costs (light purple = R, dark purple = L) or asymmetric cost (light pink = R, dark pink = L). **D.** Posterior mean estimation under uncertainty. Stimuli were selected from one of nine equally spaced measurement distributions with varying uncertainty (left). Psychometric curves showing the proportion of rightward judgements as a function of stimulus location and uncertainty (right). **E.** Same as D but for inference with prior-induced biases.

of the decision circuit to these stimuli and analyze the resulting spiking activity of the circuit the same way an experimentalist would analyze cortical data. We furthermore quantify how these responses change with uncertainty and prior biases. All figures shown are for the decision circuit; equivalent results for the inference circuit can be found in [22].

**Signatures of uncertainty in single neurons**

We quantified a range of response statistics that are commonly used to characterize spiking activity in the cortex using simulated data from the model. For each stimulus condition, we further varied posterior uncertainty between three levels, from low to high uncertainty (analogous to changing from high to low contrast in V1 experiments) (Fig. 4A).

Measuring the tuning function of individual neurons revealed that they decrease their peak firing and increase their width as uncertainty increases (Fig.4B). This modulation differs from the behavior of the inference module [22] or from data in early visual cortex [30]. The difference can be easily understood in the context of our unified coding scheme. In both modules, individual neurons are responsible for particular directions in signal space (here signal refers to either $\mathbf{x}_t$ or $\mathbf{c}_t$), defined by the corresponding columns of the decoding matrix. They will increase their firing whenever the underlying signal is in their preferred region of stimulus space. In the inference module, this means that the average firing rate of the neuron will reflect the posterior mean. In the decision module, neurons inherit their stimulus selectivity from their preferred $\mathbf{c}$ axis. As uncertainty increases, their preferred $\mathbf{c}_k$ will reduce its amplitude, triggering a corresponding reduction in the neuron's firing for a preferred stimulus. Likewise, $\mathbf{c}_k$s far away from the stimulus, which were originally inactive, will start to participate in the signal; their corresponding neurons will also increase their firing rates, which leads to a widening of their tuning function. Putting together the different effects, as a function of similarity of the neuron's preferred stimulus with the presented stimulus, we find reductions in firing rates for the preferred stimulus and an increase in firing away from the preferred stimulus, with the magnitude of the effects scaled by uncertainty (Fig. 4C).

In contrast, the effects of uncertainty on variability were consistent across circuits (see corresponding figure in Ref.[22]), with mean Fano factors close to 1 in the low uncertainty conditions and systematic

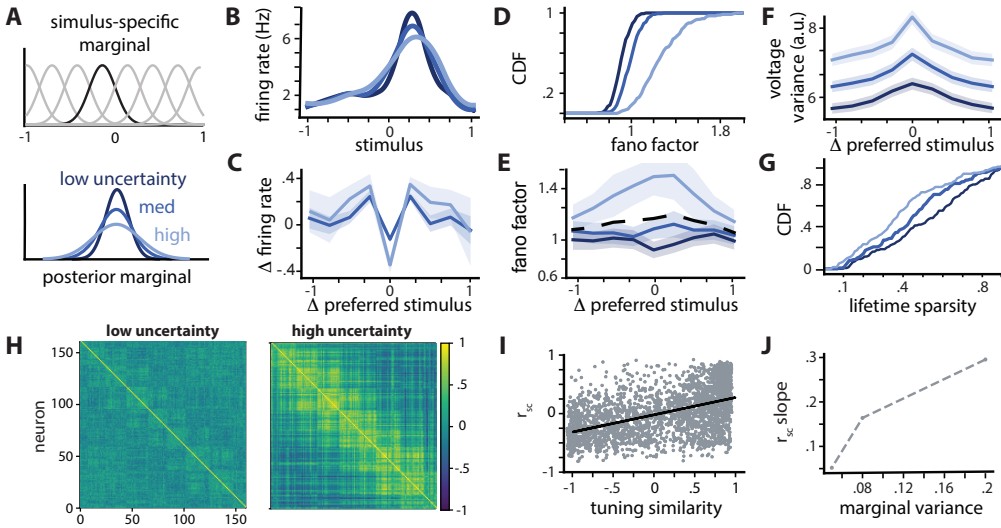

Figure 4: **Modulation of single neuron responses by uncertainty. A.** Simulated experiment with a range of stimuli that induce marginals with different means and the same variance; the uncertainty manipulations change their width. **B.** Tuning curves for a representative example neuron as uncertainty changes (with spline smoothing). **C.** Fractional changes in firing rate relative to the high certainty condition as a function of the distance between the stimulus and the neuron's preferred stimulus. **D.** Fano factor cumulative distributions. **E.** Average Fano factors and **F.** Voltage variance as a function of the stimulus similarity to the neuron's preferred stimulus. Black dashed line in E. shows average across all conditions. **G.** Lifetime sparsity cumulative distributions. **H.** Noise correlations between neuron pairs; neurons sorted by stimulus preference . **I.** Noise correlations as a function of tuning similarity; linear regression fit in black. **J.** Slope of regression line as a function of marginal uncertainty. All summary statistics were measured using 20 trials per stimulus condition.

rightward shifts of the Fano factor distribution with increasing uncertainty (Fig. 4D). This property can again be understood in the context of the encoding scheme: the variability of individual neurons can be decomposed into Poisson variability due to the encoding scheme [24] and signal variability. The first is largely unchanged across manipulations, but the increase in uncertainty leads to proportional increases in the signal — directly in the case of the inference circuit and indirectly in the decision module, since the variance of an average estimate also changes with the variance of the random variable being averages. Similar Fano factor increases are also a robust feature of experimental data [31, 17, 32].

Recent work also showed that average Fano factors over a range of uncertainties were roughly constant across stimulus orientations, both in a sampling-based model of V1 activity [17] and in data from awake macaque V1 [32]. While our model recapitulates this behavior on average (Fig. 4E, black dashed line), Fano factors start to increase at the preferred orientation for high uncertainty stimuli, a novel prediction for future data analyses. We found a similar pattern in the voltage variance (see detailed model description in Suppl.Info.), which increases with uncertainty and peaks at the preferred orientation (Fig. 4F), in agreement with previous experimental results [17, 33] as well as previous distributed sampling theory [34]. We also found that the sparsity of neural responses decreases with uncertainty, similar to that in the inference module (Fig. 4G).

Lastly, we analyzed the properties of pairwise correlations between neurons in the same setup. When visualizing the matrix of noise correlations between neurons (measured by Pearson correlations between their instantaneous firing rates over the course of a trial), with neurons indexed by their preferred stimulus, we found its structure to be modulated by uncertainty, with correlations expanding across wider stimulus ranges in high uncertainty conditions (Fig. 4H). While this observation is less intuitively explainable at the mechanistic level, it is likely to reflect the expansion of the active $c_k$ range, possibly paired with the competitive process that ensures the normalization of the approximate marginal distribution. When organizing the noise correlations by the tuning preference of the two neurons, we found that similarly tuned neurons have higher correlations than neurons with different

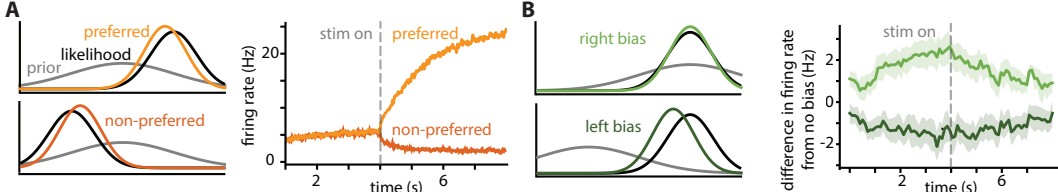

Figure 5: **Signatures of internal evidence integration.** Distributions in color represent posterior marginal. **A.** Left: a neuron's perspective on probabilistic decision making comparing a scenario when the posterior mean is close to its preferred stimulus (top) vs. the posterior attributing high probability to non-preferred stimuli (bottom). Right: Evolution of average firing rates over time for the two scenarios. **B.** Prior manipulations (left) and the change in average neuron activity relative to no bias, for the two conditions (right).

tuning preferences (Fig. 4I), as seen experimentally [35]. Moreover, this dependence increases in strength with uncertainty (Fig. 4J), as previously reported for the inference circuit [22], leading to a robust prediction for this kind of sampling based computation, which can be tested experimentally.

**Signatures of internal evidence integration**    So far, we have investigated the neural responses of the second module through the lens of statistics commonly used to characterize sensory neuron activity. Here, we analyze the same responses as one would for a decision circuit, such as LIP.

In particular, we focused on the unfolding of single neuron responses over time after stimulus onset (Fig. 5A), segregating the neurons into subsets whose tuning aligns with the presented stimulus (light pink) or not (dark pink). We found that neurons tuned to the stimulus ramp up their responses over time, as a reflection of the approximate posterior accumulating more posterior samples and thus increasing the corresponding $c_k$ values. In contrast, neurons tuned to other stimuli decrease their responses, as the posterior density associated to that stimulus region decreases. The amplitude of these ramps depends on the overall uncertainty level, such that firing rates saturate at higher levels for higher certainty stimuli when the stimulus is in the preferred direction, similar to LIP responses found by Roitman and Shadlen [36] during sensory evidence integration. The key distinction is that here we are integrating *internal* evidence (a stream of samples) rather than external evidence (a stream of noisy stimuli). Or, put another way, the model we're proposing computes a distribution over the decision variable and not a point estimate of the decision variable itself, as is the case in traditional evidence integration. What has been taken to be a key signature of evidence integration (firing rates increasing as more evidence is shown over time) is also consistent with our probabilistic coding scheme, despite the different computational goals.

The second manipulation we consider is one of the prior, inspired by an experiment by Rao and collaborators [37], in which the prior mean shifts rightwards or leftwards relative to a 'no bias' condition in which the prior is uninformative (Fig. 5B). After isolating the subset of neurons activated in the no bias condition, we compared their responses to the 'right bias' condition (light green) in which the prior aligns with the likelihood slightly sharpening the posterior without significantly changing its mean, and 'left bias' (dark green), which shifts the posterior away from the monitored neural population. Before the stimulus is presented, the neurons increase their activity when encoding the right bias, when the $c_k$s that these neurons are tuned to have high values, and similarly decrease their firing for the left bias. However, after the stimulus is turned on, this discrepancy slowly decreases: as the likelihood is strongly informative, the differences between posteriors are much less prominent than the differences between the two priors, so as more samples are integrated, the difference in responses are reduced so as to reflect the relatively minor difference between the likelihood and both posteriors. Again, LIP is known to show similar responses, where the effect of prior biases diminishes as the circuit accumulates more sensory evidence [37].

**Signatures of probabilistic computation at the population level**

The encoding of information in our model is naturally distributed across entire spiking neural networks and allows us to find robust signatures of the underlying probabilistic computation in the form of neural trajectories in low-dimensional manifolds [22]. If the map from neural responses to the

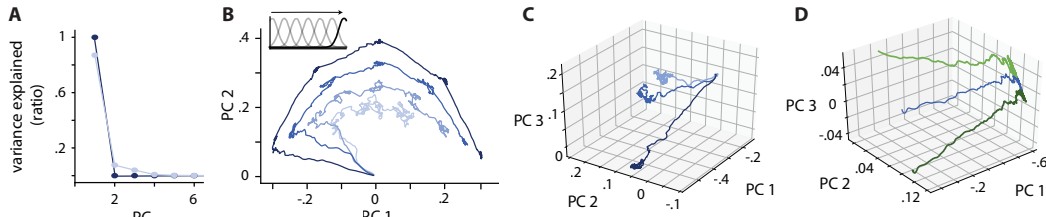

Figure 6: **Latent population dynamics. A.** Variance explained by increasing size of latent manifold for low (dark blue) and high (light blue) uncertainty; estimates use trial-average PCA for responses to 9 probe stimuli. **B.** Latent dynamics of the decision circuit in response to sequentially presenting the test stimuli (inset), for different uncertainty levels (colors). **C.** Latent dynamics of three presentations of the same stimulus, for different uncertainty conditions. **D.** Latent dynamics for three presentations of the same stimulus with an uninformative prior (blue) or left/right prior biases (green); initial conditions differ by prior, the likelihood is the same across conditions.

embedding manifold is easy to understand for the sampling circuit (the manifold maps one-to-one to the latent axes of the encoded posterior and the circuit dynamics to MCMC samples from it), this is not necessarily the case for the decision circuit. In particular the effective dimensionality of this circuit may change depending on uncertainty level. The intuition for this is simple: if the test stimuli are relatively sparse and the posteriors sharp, then a fraction of the parameters $c_k$s will not be recruited by any of the test trials. These 'collapsed' axes reduce the effective dimensionality of the dynamics; increasing uncertainty recruits more of these parameters, increasing effective dimensionality. Paired with the fact that $c_k$s are strongly correlated, dynamics appear almost invariably one dimensional in our toy examples (Fig. 6A), as seen in LIP data [38].

We can look at the low dimensional embedding of these dynamics for signatures of both uncertainty and prior biases. For instance, we simulated a scenario where the model is presented with the full sequence of test stimuli (from left to right) at different uncertainty levels, and traced the corresponding decision circuit trajectories in the space of their first two principal components (Fig. 6B), similar to [39] . This revealed structured trajectories whose geometry directly reflects posterior uncertainty. We further probed the effect of uncertainty on single stimuli (Fig. 6C). Here, the trajectories start at the same point (a location defined by the prior distribution) and gradually separate out to reflect different posteriors (Fig. 6C). The stimulus with the highest uncertainty (light blue) is closer to the prior, which nicely mirrors the observation that with limited evidence a Bayesian observer depends more on prior knowledge. The same pattern was seen in monkeys performing a time interval reproduction task [40]. The variance around the specific fixed point also depends on uncertainty, with more variability for higher uncertainty, analogous to (and inherited from) the inference circuit.

Finally we also considered a scenario where the likelihood is fixed, but the prior varies from leftward biased (dark green) to uninformative (blue) and rightward biased (light green). In this scenario, all three trajectories start in different regions of phase space (reflecting their respective priors) but quickly converge to roughly the same region of phase space (since the likelihood is narrow and dominates posterior estimates), with slight offsets that reflect the bias. Overall, we found that the decision circuit dynamics are low-dimensional and finely structured to reflect aspects of probabilistic computation, in a very different way from the sampling trajectory that drives them.

**Inference under changing conditions**

The online nature of the marginalization procedure implemented by the decision circuit allows it to integrate samples from any posterior, static or dynamic, within a time window specified by parameter $\alpha$. This will lead to dependencies over time in the outputs of this circuit within the time horizon of integration, with possibly measurable behavioral signatures. Although the current implementation of the distributed sampling network only considers inference with static stimuli, we can get a sense for how the coding scheme is affected by stimulus dynamics by simulating a slightly more complex setup, with a succession of two static stimuli.

Figure 7A shows a simple simulation of a changing distribution, where the sensory input switches halfway through the simulation such that the mean of the marginal posterior changes. The decoded

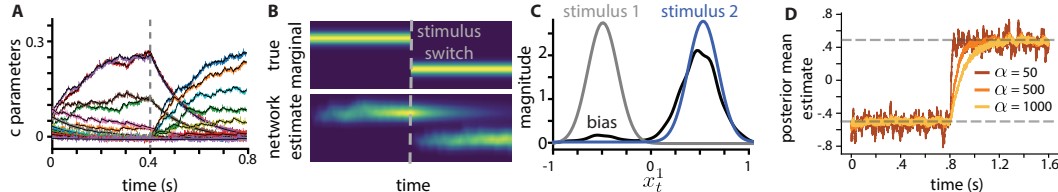

Figure 7: **Switching between two static stimuli. A.** Dynamics of decoded parameters **c** when the stimulus switches 400ms (dashed line) into the simulated trial. **B.** The evolution of the decoded marginal over time. **C.** The decoded marginal distribution (black) 400ms after the stimulus switch. **D.** Decoded posterior mean estimate over time, for different values of integration time constant $\alpha$. Dashed lines show ground truth for the two stimuli.

marginal distribution shows lingering traces of the previous stimulus, with the approximate marginal exhibiting bimodality (Fig.7B) that still persists 400ms after the switch (Fig.7C). Our model predicts that the strength of this bias should be related to the uncertainty of the preceding stimuli, and the similarity between the stimuli (formally, the distance between their corresponding posterior means). The temporal extent of this bias is controlled by the leak, or evidence discount rate, of the marginal posterior (Fig.7D). This is reminiscent of the history dependence of choice biases seen in sequential decision making [41, 42]. It may even be beneficial for dynamic inference, since exponential discounting of past evidence can be used to approximate Bayes-optimal inference in non-stationary environments [43]. It leads however to an interesting conundrum: on the one side, a precise representation of the marginal requires a large $\alpha$ so as to integrate as many samples as possible; on the other side, $\alpha$ should be small if we are to minimize sequential biases. The optimal trade-off between the two is dictated by the time scale of changes in the the sensory stimuli. Indeed, discount rates are known to reflect the expected rate of change of the statistics of the environment in a similar manner [44]. We expect that the integration time constant $\alpha$ is fixed within a given experimental context, but may adapt over time to reflect the time constants of environmental stability. The $\alpha$ parameter can in principle be estimated independently at the level of behavior and in neural data, which could be used to generate experimental predictions of the theory.

## 3    Discussion

Although behavioral evidence supports the idea that humans and animals use uncertainty to guide nearly optimal behavior (but see [45–47] for counter-arguments), the neural underpinnings of probabilistic computation remain controversial [15, 6, 48]. Contrary to the tradition of viewing parametric and sampling-based codes as mutually exclusive, here we use them both as useful data structures, needed at different computational stages for approximately optimal decision making. First, sensory circuits use sampling to represent the joint statistics over many features and to marginalize out nuisance variables, which is hard with parametric representations [49] . Second, in decision areas, parametric codes provide a compact, quasi-instantaneous representation of the marginals which can be easily combined with costs to yield approximately optimal decisions. Thus, our model provides a computationally well-justified reconciliation between competing probabilistic neural codes, which is complementary to recent attempts at unifying parametric and sampling-based models through the lens of Bayesian encoding and decoding [50].

Our decision circuit could in principle be adapted to represent a two-dimensional marginal posterior using two-dimensional kernel functions. The corresponding circuit size would need to grow linearly in the number of parameters, $\vec{c}$, and in principle exponentially with the number of marginal posterior dimensions. However, higher dimensional marginals may not be necessary: one should be able to include the decision variable explicitly as part of sampling-based hierarchical inference, making the relevant marginals always 1-D. These marginals could, in principle, change to adapt to different tasks using any of the biologically plausible proposals for dynamic information routing between neural circuits [51–53].

Our model involves two stages of computation and relies on the same spike-based framework [24] to encode the corresponding probabilistic quantities. This shared encoding means that many classic measures of responses in single cell and neuron pairs are preserved across processing stages and

largely consistent with those reported for early sensory cortical responses. Furthermore, the points of difference match experimental points of contention between parametric and sampling-based codes (e.g. the modulation of tuning functions by uncertainty) [9, 22]. Finally, although the idea that neural computation relies on coordinated activity between neurons is central to the construction of both circuits, they significantly diverge in the properties of their latent trajectories. Specifically, the two circuits have different slow dynamics that reflect the different underlying computations, leading to diverging experimental predictions. Uncertainty manifests mainly in the entropy of latent trajectories for sampling [22], whereas in the decision circuit it affects amplitude of mostly deterministic trajectories. These different signatures point to approaches based on latent dynamical systems models [54, 55] as a way to potentially disambiguate between different stages of probabilistic computation.

Our decision circuit can be thought of as a probabilistic form of evidence integration akin to that proposed by Boerlin and Deneve [25], with the 'evidence' given by samples from the network's internal model rather than noisy external inputs. This close analogy prompts the idea that from the perspective of a brain trying to act in the world —or our decision circuit — there may be little point in making a distinction between internal and external noise [56]. This significantly complicates the experimental validation of neural sampling in a way that can only be addressed by further theoretical work. To explore such complex scenarios, a natural next step is to expand the first module from inference with static stimuli to inference based on continuous stimulus streams with nontrivial temporal correlations. This could in principle be achieved by using particle filtering as underlying sampling dynamics [57], with particles embedded in the same spiking recurrent circuit (as previously demonstrated for multiple parallel chains sampling from the same static posterior [22]). Exploring the interaction between such inference dynamics and our probabilistic decision circuit could reveal new neural and behavioral signatures of sampling-based probabilistic computation in the brain.

## Acknowledgments and Disclosure of Funding

We thank Edoardo Balzani, Pedro Herrero-Vidal and Colin Bredenberg for helpful discussions and feedback on earlier versions of this manuscript. CRB is supported by the National Science Foundation Graduate Research Fellowship under Grant No. DGE1839302. CS is supported by National Institute of Mental Health Award 1R01MH125571-01, by the National Science Foundation under NSF Award No.1922658 and a Google faculty award.

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
