# Supplementary Information: A sampling-based circuit for optimal decision making

**Camille E. Rullán Buxó**
Center for Neural Science
New York University
New York, NY 10003
ch2880@nyu.edu

**Cristina Savin**
Center for Neural Science
Center for Data Science
New York University
New York, NY 10003
csavin@nyu.edu

## 1 Inference in a linear Gaussian model

The generative model for our observations is defined as $\mathbf{s} = \mathbf{A}\mathbf{x} + \boldsymbol{\varepsilon}$. We assume that the latent $\mathbf{x}$ is normally distributed, $\mathcal{N}(\boldsymbol{\mu_x}, \boldsymbol{\Sigma_x})$, and that it has dimensionality $D$. We further assume that the observation is generated by a linear mapping of $\mathbf{x}$ through a matrix $\mathbf{A}$ with dimensionality $M \times D, M > D$. This measurement is corrupted by i.i.d. noise drawn from a zero mean Gaussian, $\boldsymbol{\varepsilon} \sim \mathcal{N}(0, \sigma^2 \mathbb{I})$. The observation model is $P(\mathbf{s}|\mathbf{x}) = \mathcal{N}(\mathbf{A}\mathbf{x}, \mathbf{A}\boldsymbol{\Sigma_x}\mathbf{A}^\top + \boldsymbol{\Sigma_\epsilon})$, and the posterior is a Gaussian distribution with mean

$$\boldsymbol{\mu_{x|s}} = (\mathbf{A}\boldsymbol{\Sigma_x})^\top (\mathbf{A}\boldsymbol{\Sigma_x}\mathbf{A}^\top + \sigma^2 \mathbb{I})^{-1}\mathbf{s} \tag{1}$$

and variance

$$\boldsymbol{\Sigma_{x|s}} = \boldsymbol{\Sigma_x} - (\mathbf{A}\boldsymbol{\Sigma_x})^\top (\mathbf{A}\boldsymbol{\Sigma_x}\mathbf{A}^\top + \sigma^2 \mathbb{I})^{-1}\mathbf{A}\boldsymbol{\Sigma_x} \tag{2}$$

The inference circuit samples this posterior using simple Langevin sampling dynamics as in [1].

## 2 Network implementation

### 2.1 Linear dynamics

The network implements dynamics defined by

$$\dot{\mathbf{c}}_t = \frac{1}{\alpha}(f(\mathbf{c}_{t-1}) + \boldsymbol{\phi}(x_t)) \tag{3}$$

where vector $\boldsymbol{\phi}(x_t)$ concatenates all basis $K$ functions evaluated at the current input. At the fixed point, $\dot{\mathbf{c}}_t = 0$, and $f(\mathbf{c}_{t-1}) = \boldsymbol{\phi}(x_{it})$. We assume that $f(\mathbf{c}_{t-1}) = \beta\mathbf{c}_{t-1}$. In order to ensure that the fixed point is a properly normalized distribution, $\sum_k \phi_k(x_t) = \sum_k \beta c_{k,t} = 1$. We therefore set $\beta$ to $K\bar{\phi}$, where $\bar{\phi}$ is the average magnitude of the kernel functions.

For all of our simulations, the kernels were a set of 20 Gaussians with $\sigma = 0.06$, centered such that they evenly tile the range from $-1$ to $1$, and $\beta = 9.45$.

### 2.2 Decision making circuit dynamics

The framework that we use to implement the marginalization circuit is a stochastic version of the the balanced spiking network (BSN) by Boerlin et al [2, 3]. Briefly, this framework proposes a way of embedding a linear dynamical system defined by (3) in the spiking activity of a network of $N$ neurons. The network's estimate of the desired dynamics, $\hat{\mathbf{c}}_t$, can be read out linearly from the filtered spike trains $\mathbf{r}_t$ through a set of $K \times N$ readout weights, $\mathbf{D}$. The $i$'th component of the vector $\mathbf{r}_t$ is

35th Conference on Neural Information Processing Systems (NeurIPS 2021), .

given by $r_t^i = o_t^i * h_t = \int_0^t e^{-t'/\tau_d} s_{t'}^i dt'$ where $o_t^i = \sum_{t_{sp}^i} \delta(t - t_{sp}^i)$ denotes the $i$'th neuron's spike train, defined by a series of delta functions at spike times $\{t_{sp}^i\}$, and $\tau_d$ is the time constant of the exponential filter $h_t$.

By postulating that a neuron spikes to reduce the mean squared error between the true dynamics, $\mathbf{c}_t$, and the network's estimate of the dynamics, $\hat{\mathbf{c}}_t$, we can derive a spiking condition where the voltage of a neuron is proportional to the residual between $\mathbf{c}_t$ and $\hat{\mathbf{c}}_t$, and the neuron spikes whenever the error exceeds a certain value proportional to its corresponding decoding weight, $\mathbf{D_i}$.

Specifically, the voltage update equation for our network can be written as

$$v_t^i = v_{t-1}^i + \mathbf{D_i}^\top \left( \left( \frac{\beta}{\alpha} + \frac{1}{\tau_d} \right) \mathbf{D} r_{t-1}^i + \frac{1}{\alpha} \phi \left( x_t^i \right) \right) dt \tag{4}$$

The Poisson BSN proposed in [3] introduces a soft threshold for spiking such that spike probability grows as a nonlinear function of membrane potential. Specifically, each neuron's conditional intensity function is a sigmoidal function of its membrane potential:

$$\lambda_t^i = f(v_t^i) = \frac{F_{max} - F_{min}}{1 + e^{-\gamma(v_t^i - T)}} + F_{min} \tag{5}$$

where $v_t^i$ is the membrane potential at time $t$, $T$ is the spike threshold, $\gamma$ is a slope parameter governing the sharpness of the threshold, $F_{max}$ is the maximal firing rate and $F_{min}$ is a baseline firing rate, meant to simulate random firing activity in the absence of a stimulus. The spike threshold is equal to $\frac{1}{2}||\mathbf{D_i}||_2^2$, as in [2]. For each time bin, neuron $i$ spikes with a probability equal to

$$P \left( o_t^i = 1 | \lambda_t^i \right) = 1 - \exp(-\Delta \lambda_t^i) \tag{6}$$

After each spike, $o_t^i = 1$, the filtered spike trains $\mathbf{r}_t$ are augmented and the membrane potential is reset:

$$\mathbf{r}_t = \mathbf{r}_t + \mathbf{o}_t \tag{7}$$

$$\mathbf{v}_t = \mathbf{v}_t - \mathbf{D}^\top \mathbf{D} \mathbf{o}_t \tag{8}$$

which ensures that post-spike membrane potential equals the difference between the target variable, $\mathbf{c}_t$ and network output, $\hat{\mathbf{c}}_t = \mathbf{D} \mathbf{r}_t$.

The full derivation of the Poisson-noise embedding network can be found in [3].

## 3 Simulations

### 3.1 Simulation parameters

Unless otherwise specified, the network parameters for all of our simulations were:

- $N$ (number of neurons) = 160
- $\tau_d$ (spike rate time constant) = 20
- $F_{max}$ (maximal firing rate) = 1
- $F_{min}$ (minimal firing rate) = 0
- $\gamma$ (slope of spiking threshold nonlinearity) = $10^5$
- $\alpha$ (decay rate of kernel dynamics) = 800
- $dt$ = 0.1 ms

The decoding weights were randomly initialized from a standard normal distribution and re-scaled such that the magnitude of each neuron's decoding vector in all dimensions was 0.1.

Fig. 2 shows a simple demonstration of the decision circuit using samples from a 2D posterior generated by the inference circuit. Code to implement the simulation can be found on `https://github.com/camillerb/RullanSavin2021`. In this case, the inference circuit is sampling from the posterior of a linear Gaussian model. The prior is a Gaussian with zero mean and variance

$\begin{bmatrix} 0.8 & -0.3 \\ -0.3 & 0.2 \end{bmatrix}$, the observation matrix is the identity matrix and the measurement noise has a variance of $\sigma = 0.1$ for each dimension. The measurement, $s$, was placed at $\begin{bmatrix} -0.2 \\ 0.1 \end{bmatrix}$. The full posterior is then a Gaussian distribution with mean $\mu_{x|s} = \begin{bmatrix} -0.18 \\ 0.08 \end{bmatrix}$ and variance $\Sigma_{x|s} = \begin{bmatrix} 0.08 & -0.02 \\ -0.02 & 0.05 \end{bmatrix}$. The second network encodes a marginal over the first dimension, or a one-dimensional Gaussian with mean $\mu_{x_1|s_1} = -0.18$ and variance $\sigma_{x_1|s_1} = 0.08$.

The bi-modal posterior in Fig. 3A-C is a mixture of two Gaussians with means $\mu_1 = -0.5$ and $\mu_2 = 0.75$ and variances $\sigma_1 = 0.3$ and $\sigma_2 = 0.15$, with weighting $w_1 = 0.6$ and $w_2 = 0.4$. The MAP estimate is $0.75$ and the posterior mean is $0.125$, but strictly speaking there is more probability mass on the left side of the boundary (probability mass ratio $= 0.98$). In the case of asymmetric cost, option 'L' has an expected cost of $1.25$ and option 'R,' $0.5$.

Fig. 3D was generated by drawing a stimulus from a measurement distribution centered at one of nine locations, evenly spaced between $-1$ and $1$. The prior was a Gaussian with zero mean and variance $\sigma = 0.8$ and the measurement noise was varied between high uncertainty ($\sigma_H = 0.5$), medium uncertainty ($\sigma_M = 0.3$), and low uncertainty ($\sigma_L = 0.15$). Each simulation was repeated twenty times, with the same stimulus location but different samples from the posterior. For Fig. 3E, the same procedure was repeated but with a prior centered at $-1$ or $1$.

For Fig. 4, we analyzed the network output from 25 repetitions of simulations with measurements at one of eleven evenly spaced locations between $-1$ and $1$ with the same prior as Fig. 3. The measurement noise was $\sigma_H = 0.8$, $\sigma_M = 0.4$, and $\sigma_L = 0.2$. Each simulation was run for 8s (8000 time bins).

Fig. 5A was generated using the same broad prior and stimuli used for Fig. 4. Neurons were sorted by their preference to stimuli, with the stimuli eliciting the highest response categorized as its preferred stimuli, and the one eliciting the lowest response as its non-preferred stimuli. The firing rate responses shown were averaged over 10ms time windows, then over all neurons. The measurement noise was $\sigma = 0.1$.

For Fig. 5B, we similarly calculated the firing rate time courses of the 50 most active neurons when presented with a stimulus at $0.5$ with measurement noise of $\sigma = 0.1$ and a prior centered at zero. We then subtracted that from the firing rate time courses for the same neurons in the case of a leftward ($\mu_x = -0.5$) or rightward ($\mu_x = 0.5$) biased prior.

To generate Fig. 6B, we continuously fed samples from a posterior centered at one of five evenly spaced locations between $-1$ and $1$. The measurement distributions had widths of $\sigma = 0.4, 0.2, 0.1$ and $0.05$, and the prior was once again zero mean and variance $\sigma = 0.8$. Fig. 6A shows the variance explained for $\sigma = 0.4$ (lightest blue) and $\sigma = 0.05$ (darkest blue). Figs. 6C-D were generated using the neural activity from the simulations in Fig. 3D-E.

Finally, for Fig. 7, the network received 4000 samples first from a posterior centered at $-0.5$ and then the same amount of samples from a posterior centered at $0.5$ with observation noise $\sigma = 0.15$. For Fig. 7A-C, the simulation shown has $\alpha = 500$.