# OpenReview forum: "A sampling-based circuit for optimal decision making"
_NeurIPS.cc/2021/Conference — NeurIPS 2021 Spotlight_

### Official Review · Reviewer_1gKY · 2021-07-01

**Rating:** 8
**Confidence:** 4

**Summary:**

This is a clearly written and interesting paper that develops a model integrating sample-based inference into perceptual decision making. Given the prevalent interest in both topics, their integration will be potentially impactful.

**Limitations And Societal Impact:**

Yes.

**Main Review:**

The main issue for me is that it wasn't clear to me which phenomena are predicted by which models. The authors discuss several alternative models early on, and occasionally mention them later in the paper, but these comparison are haphazard. What's needed is something like a table that presents the full cross-section of models and critical phenomena that discriminate between them. It doesn't need to literally be a table, but that conveys the idea.

Other comments:

p. 2: It is stated that parametric representations are needed for combining a task-specific cost function with the marginal distribution to produce a decision. I don't understand why a nonparametric representation can't be used here. Why can't you do decision theory with samples?

I thought it was interesting that the model could produce a form of serial dependence, which has been extensively documented in studies of perceptual decision making. Much is known about the factors controlling the strength of serial dependence (see for example studies by Urai, Donner, and others). The paper doesn't include anything on this, which seems like a missed opportunity.

**Time Spent Reviewing:**

2.5 hours

---

> ### Author Response · Authors · 2021-08-09
> **Reply to Reviewer 1gKY**
>
> We agree that there is a lot of confusion in the literature about what exactly the different probabilistic coding models predict and how they differ. We will do our best to clear up that part of the text in the introduction and discussion. However, fully organizing the literature across these axes would be better suited for a review.
>
> One can perform statistically optimal decisions with samples, but at some point the samples (which may be spread out over time and/or over a population of neurons) need to be converted into a spatially and temporally localized representation. This is precisely what the decision circuit does. Our work is the first, but by no means the only, proposal for how to do this.
>
> Finally, thank for these references. We will be incorporating those into the discussion, particularly the 2019 study by Urai et al. showing the effect of past choices on evidence accumulation - which may be related to the bias we observe in figure 7.

---

> > ### Comment · Reviewer_1gKY · 2021-08-25
> > **response**
> >
> > Thanks for the clarifications. Very nice paper.

---

### Official Review · Reviewer_sRma · 2021-07-15

**Rating:** 7
**Confidence:** 3

**Summary:**

This study develops a two-layer spiking neural network to achieve decision making. In particular, the network model has a hybrid coding scheme in two layers, as an attempt to bring different neural coding paradigms into a coherent model. The 1st layer uses a sampling-based code to infer a posterior distribution and marginalize nuisance variable, and the 2nd layer uses a parametric code based on kernel density estimation to achieve decision making with a task-specific cost function. The paper also presents neural activity statistics and low-dimensional population dynamics in order to find the neural signature of sampling-based decision making and compare it with experimental data.

**Limitations And Societal Impact:**

Yes

**Main Review:**

A contribution of this study is that through properly combining different neural codes the whole network can implement computations that cannot be easily achieved by either code, since different neural codes have different advantages in implementing some computation. This is very important to the neural coding field by unifying different neural codes together.

### Major comments:
- The two layers in the network are implemented in the same spiking neural network (ref. 24), if I understood correctly. Then I have a conceptual question that how the same network adopts completely different neural codes, and achieve different computations? Do the 1st and 2nd networks have different parameters and are set into different dynamic regimes? I haven’t seen there is any discussion about this across the whole paper.

- The parameter c’s dynamics in the 2nd circuit (eq. 1) by design is letting it finally converge to a stationary state which normalizes the distribution p(x_i). Later in Fig. 5 the paper shows the 2nd circuit is able to accumulate evidence over time. I am wondering how evidence accumulation could be implemented in c’s dynamics which has different original designing purposes.

I hope the authors could discuss these two issues in the rebuttal and in a revised manuscript.

### Clarity
The writing of the whole paper is structure-wise. And the neural signature further consolidates the biological plausibility of this model. But when I read the paper I sometimes got confused that whether the presented neuronal responses are from the 1st or the 2nd circuit.

From the neuroscience point of view, the writing of the paper is a little abstract.
- For example, the connections from the 1st circuit to the 2nd circuit is defined by the Gaussian kernels as a function of the low-dimensional projection x_i from the 1st circuit. How the feedforward connections will look like from neurons in the 1st circuit to neurons in the 2nd circuit?
- It is unclear how c could be read out from neuronal responses in the 2nd circuit? And how the decoding weight of c is determined.

### Minor
- Does all the network responds to a fixed s randomly sampled from the generative model? Is all the fluctuation of responses of 2nd circuit coming from the sampling-based inference?
- Is there any noise in the 2nd circuit? Or the fluctuation of the 2nd circuit is completely inherited from the sampling variability in the 1st circuit?
- Typo in Figure 3d: the light blue curve should be low contrast if I understood correctly.
- What do the light/dark colors in Figure 3d mean? Left or right choice?


**Time Spent Reviewing:**

3

---

> ### Author Response · Authors · 2021-08-09
> **Reply to Reviewer sRma**
>
> The encoding procedure that maps the latent dynamical system into spiking activity is the same for both circuits, but the two circuit have different slow dynamics that reflect different underlying computations (inference or marginalization);  since the properties of the spiking neural activity reflect these distinct latent dynamical systems, they lead to different experimental predictions w.r.t. spiking neural activity.
>
> The model we’re proposing is an end-to-end probabilistic representation, where the quantity being computed is a distribution over the decision variable and not the decision variable itself, as is the case in traditional evidence integration. Although the input to a neural circuit doing either of these computations would be similar, the key computational dissimilarity is that we encode a posterior, not a point estimate. In other words, traditionally external stimuli are taken as evidence in favor of one hypothesis over another. Here, we are accumulating samples, or “internal” evidence, which come from inverting the generative model for the stimuli. In figure 5, we show that what has been taken to be a key signature of evidence integration (firing rates increasing to a bound as more evidence is shown over time) is also consistent with our coding scheme, despite the slightly different computational goals.
>
> The key neural analyses are performed on the second circuit, and we will edit the text for clarity.
>
> The feed-forward connections from the first circuit to the second involve local non-linearities which could be implemented by dendrites.
>
> The decoding of c is linear, with the decoder weights pre-specified, consistent with the encoding framework by Deneve et al; there exist extensions suggesting how the weights could be learned through local plasticity, see eg. Brendel et al. (2020).
>
> We are implementing inference in a static generative model (although we do show how the network latent dynamics respond to abrupt changes in stimuli in figure 6). The fluctuations in the 2nd circuit are inherited from sampling.
>
> Finally, thank you for spotting the typo, we have corrected that mistake. The light color indicates low contrast and the dark color indicates high contrast. Choice is shown in the right panel, where we are reporting the proportion of times the network outputs a rightward choice.

---

### Official Review · Reviewer_Wkwk · 2021-07-16

**Rating:** 8
**Confidence:** 5

**Summary:**

How populations of neurons can represent probability distributions and perform probabilistic computations is an important question in computational neuroscience. This work proposes a spiking network model that combines the advantages of sampling-based methods and parametric representations of uncertainty, thus bringing together two different perspectives on probabilistic computation in the brain.

Sampling-based models have the appeal of being able to represent complex multivariate distributions and effectively perform inferential computations such as marginalization. However, with sampling-based codes uncertainty is represented implicitly by the activity of an ensemble of neurons across time. This work proposes the first spiking circuit for mapping the neural samples into appropriate actions. The proposed network model has two components. The first component is the _inference circuit_ which essentially implements Langevin sampling. The neural samples for a task-relevant marginal from this circuit are then fed to the _decision circuit_. The population activity of the decision circuit is an embedding of the parameters of a parametric approximation of the marginal.

Importantly, the authors also analyze the activity of the network both at the single neuron level and the population level. In addition to reproducing a variety of empirical observations, they provide concrete predictions that can be experimentally tested.

**Limitations And Societal Impact:**

There are no other potential negative societal impacts or ethical concerns.

**Main Review:**

__Significance__: This work presents a spiking network model that (i) performs probabilistic inference and (ii) constructs a parametric representation from inferred marginal samples that can be used for downstream computations. More importantly, this work provides concrete testable predictions about how uncertainty in the input and prior biases affect the population dynamics. If neuroscience experiments could validate even a few of the predictions in this work, it would be a significant step towards understanding how the brain implements probabilistic computations.

__Originality__: Several sampling-based and parametric representations of uncertainty can be found in the computational neuroscience literature. The novelty of this work is in constructing a spiking network model that combines the advantages of both these approaches.

__Quality__: In addition to the model itself, the analysis of the network activity to look for signatures of probabilistic computation is a very important contribution of this paper. The observations at both the single neuron and the population level correspond well with experimental observations.

__Clarity__: The presentation is very clear. The schematic makes the basic concept clear, and the simulation results illustrate the working of the network well.
 Minor:
In equation 1, should there be a $-$ instead of a $+$?
Panels B and C in figure 2 are easy to interpret, but the colorbar is to be included for completeness.
Panel D in figure 3, one of the labels needs to be 'low'; labels in panel E could be useful - likelihood, prior, etc
Caption in figure 6C: latent dynamics 'of' three ..
Line 237: model is presented instead of resented :)
Line 307: there may 'be' little point

__Questions__:
* Would you require a separate decision circuit for the parametric representation of each marginal distribution? Is there a way in which this model could be adapted to simultaneously represent all marginals using a single population of neurons?
* Suppose there is a task that requires the joint marginal of two variables. Can this model be easily adapted to represent a bivariate marginal?
* Can the amount of time taken to reach a certain decision-making performance level be used to estimate $\alpha$? This could potentially be experimentally tested/fit to experimental data?

**Time Spent Reviewing:**

4-5

---

> ### Author Response · Authors · 2021-08-09
> **Reply to Reviewer Wkwk**
>
> Thank you for pointing out the typos, they have been fixed in the text and the figures have been updated to reflect your suggestion.
>
> It is in principle possible to use the same decision circuit to represent different marginals at different times. This would require to dynamically change the readout from the sampling module, using any of the biologically plausible proposals for dynamic information routing between neural circuits.
>
> The decision circuit can also be adapted to represent a two-dimensional marginal posterior, which entails using 2-D kernel functions. Note that the kernel approximation scales unfavorably in higher dimensions, so the circuit size needs to grow substantially (for the same precision: linear in the number of parameters c, and in principle exponential with the number of posterior dimensions). However,  we would argue that higher dimensional marginals are not really necessary:  one should be able to include the decision variable explicitly as part of sampling-based hierarchical inference, making the relevant marginals 1-D.
>
> Finally, we expect that the integration time constant alpha is fixed within a given experimental context,  but may adapt over time to reflect the time constants of environmental stability. The time constant of integration can be in principle estimates independently at the level of behavior and in neural data, which could be used to generate experimental predictions of the theory.

---

> > ### Comment · Reviewer_Wkwk · 2021-08-31
> > **Response**
> >
> > Thank you for the clarifications. Very interesting paper, look forward to seeing experimental work for testing the ideas presented here!

---

### Official Review · Reviewer_8Xdh · 2021-07-16

**Rating:** 8
**Confidence:** 3

**Summary:**

This work proposes to combine two main (usually competing) theories of how the brain encodes probability distributions - sampling-based and population coding. Both are used in this model of probabilistic computation. The sampling-based representation is used for potentially complex joint distributions. These feed into downstream areas, which represent simpler marginal distributions using parametric encoding. Finally, the article examines application of the parametric code for decision making.
Several experimental facts are reproduced and explained in toy examples.


**Limitations And Societal Impact:**

This is a purely theoretical work which contributes to an ongoing discussion about the nature of neural coding. Limitations are as always - a certain set of assumptions and a level of simplification that are necessary to extract useful insights. It's interesting to consider scaling such a model to real-world problems, but that's a different area of research. I can't envision any negative societal impact of having this work published.

**Main Review:**

This paper is a very pleasant read.
It's a novel combination of well-known techniques. The relevant work is adequately referenced and discussed.
The submission is technically sound and very well written. It in an interesting addition to the ongoing evaluation of the Probabilistic Brain Hypothesis.

Minor typos:
- Figure 1: I think the cost matrix (2x2?) should act on P(x), not c.
- Figure 3: B: I believe y-axis is flipped. C: update caption (colour labels). D: high -> low.
- Figure 4: G: update caption.
- L 84: for clarity, I'd suggest to remove $K\langle\phi\rangle c_t$.
- L 237: "resented"
- L 248: "from from"
- L 269: "is controlled the leak"
- L 307: "there may little point"

**Time Spent Reviewing:**

3

---

> ### Author Response · Authors · 2021-08-09
> **Reply to Reviewer 8Xdh**
>
> Thank you for your comments, we will correct the typos accordingly.  For figure 1, the cost matrix acts on the c variables as a proxy for the probability distribution, since they are the kernel magnitudes spanning the space of P(x). The mistakes in figures 3 and 4 were corrected as you suggested.

---

### Decision · Program_Chairs · 2021-09-27

**Decision:**

Accept (Spotlight)

**Comment:**

Novel, technically sound contribution to the field of computational neuroscience that proposes a combined neural model for inference and decision making. Particularly, the analysis linking the neural activity of the proposed model to experimental observations was deemed valuable by reviewers. Accept.